# ANALYZING AND EXPLOITING NARX RECURRENT NEURAL NETWORKS FOR LONG-TERM DEPENDENCIES

## ABSTRACT

Recurrent neural networks (RNNs) have achieved state-of-the-art performance on many diverse tasks, from machine translation to surgical activity recognition, yet training RNNs to capture long-term dependencies remains difficult. To date, the vast majority of successful RNN architectures alleviate this problem using nearly-additive connections between states, as introduced by long short-term memory (LSTM). We take an orthogonal approach and introduce MIST RNNs, a NARX RNN architecture that allows direct connections from the very distant past. We show that MIST RNNs 1) exhibit superior vanishing-gradient properties in comparison to LSTM and previously-proposed NARX RNNs; 2) are far more efficient than previously-proposed NARX RNN architectures, requiring even fewer computations than LSTM; and 3) improve performance substantially over LSTM and Clockwork RNNs on tasks requiring very long-term dependencies.

## 1 INTRODUCTION

Recurrent neural networks (Rumelhart et al., 1986; Werbos, 1988; Williams & Zipser, 1989) are a powerful class of neural networks that are naturally suited to modeling sequential data. For example, in recent years alone, RNNs have achieved state-of-the-art performance on tasks as diverse as machine translation (Wu et al., 2016), speech recognition (Miao et al., 2015), generative image modeling (Oord et al., 2016), and surgical activity recognition (DiPietro et al., 2016).

These successes, and the vast majority of other RNN successes, rely on a mechanism introduced by long short-term memory (Hochreiter & Schmidhuber, 1997; Gers et al., 2000), which was designed to alleviate the so called *vanishing gradient problem* (Hochreiter, 1991; Bengio et al., 1994). The problem is that gradient contributions from events at time $t - \tau$ to a loss at time $t$ diminish exponentially fast with $\tau$, thus making it extremely difficult to learn from distant events (see Figures 1 and 2). LSTM alleviates the problem using nearly-additive connections between adjacent states, which help push the base of the exponential decay toward 1. However LSTM in no way solves the problem, and in many cases still fails to learn long-term dependencies (see, e.g., (Arjovsky et al., 2016)).

NARX[1] RNNs (Lin et al., 1996) offer an orthogonal mechanism for dealing with the vanishing gradient problem, by allowing direct connections, or delays, from the distant past. However NARX RNNs have received much less attention in literature than LSTM, which we believe is for two reasons. First, as previously introduced, NARX RNNs have only a small effect on vanishing gradients, as they reduce the exponent of the decay by only a factor of $n_d$, the number of delays. Second, as previously introduced, NARX RNNs are extremely inefficient, as both parameter counts and computation counts grow by the same factor $n_d$.

In this paper, we introduce MIxed hiSTory RNNs (MIST RNNs), a new NARX RNN architecture which 1) exhibits superior vanishing-gradient properties in comparison to LSTM and previously-proposed NARX RNNs; 2) improves performance substantially over LSTM on tasks requiring very long-term dependencies; and 3) remains efficient in parameters and computation, requiring even fewer than LSTM for a fixed number of hidden units. Importantly, MIST RNNs reduce the decay's exponent by a factor of $2^{n_d - 1}$; see Figure 2.

---

[1]The acronym NARX stems from Nonlinear AutoRegressive models with eXogeneous inputs.

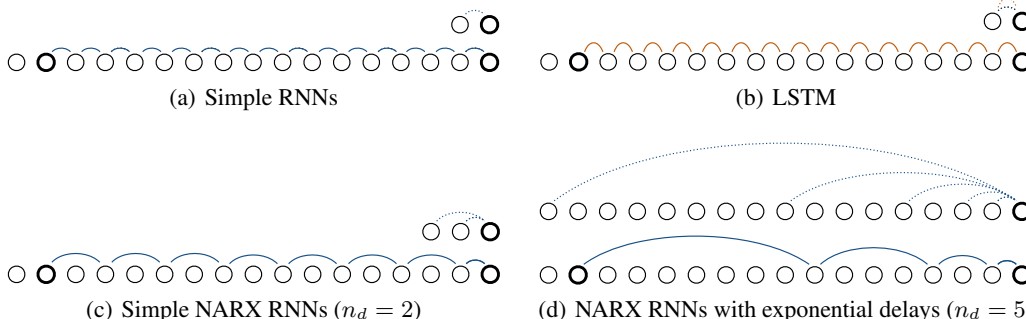

(a) Simple RNNs

(b) LSTM

(c) Simple NARX RNNs ($n_d = 2$)

(d) NARX RNNs with exponential delays ($n_d = 5$)

Figure 1: Direct connections (dashed) to a single time step $t$ and example shortest paths (solid) from time $t - \tau$ to time $t$ for various architectures. Typical RNN connections (blue) impede gradient flow through matrix multiplications and nonlinearities. LSTM facilitates gradient flow through additional paths between adjacent time steps with less resistance (orange). NARX RNNs facilitate gradient flow through additional paths that span multiple time steps.

## 2 BACKGROUND AND RELATED WORK

Recurrent neural networks, as commonly described in literature, take on the general form

$$\mathbf{h}_t = f(\mathbf{h}_{t-1}, \mathbf{x}_t, \boldsymbol{\theta}) \tag{1}$$

which compute a new state $\mathbf{h}_t$ in terms of the previous state $\mathbf{h}_{t-1}$, the current input $\mathbf{x}_t$, and some parameters $\boldsymbol{\theta}$ (which are shared over time).

One of the earliest variants, now known to be especially vulnerable to the vanishing gradient problem, is that of simple RNNs (Elman, 1990), described by

$$\mathbf{h}_t = \tanh(\mathbf{W}_h \mathbf{h}_{t-1} + \mathbf{W}_x \mathbf{x}_t + \mathbf{b}) \tag{2}$$

In this equation and elsewhere in this paper, all weight matrices $\mathbf{W}$ and biases $\mathbf{b}$ collectively form the parameters $\boldsymbol{\theta}$ to be learned, and $\tanh$ is always written explicitly[2].

Long short-term memory (Hochreiter & Schmidhuber, 1997; Gers et al., 2000), the most widely-used RNN architecture to date, was specifically introduced to address the vanishing gradient problem. The term LSTM is often overloaded; we refer to the variant with forget gates and without peephole connections, which performs similarly to more complex variants (Greff et al., 2016):

$$\mathbf{f}_t = \sigma(\mathbf{W}_{fh} \mathbf{h}_{t-1} + \mathbf{W}_{fx} \mathbf{x}_t + \mathbf{b}_f) \tag{3}$$
$$\mathbf{i}_t = \sigma(\mathbf{W}_{ih} \mathbf{h}_{t-1} + \mathbf{W}_{ix} \mathbf{x}_t + \mathbf{b}_i) \tag{4}$$
$$\mathbf{o}_t = \sigma(\mathbf{W}_{oh} \mathbf{h}_{t-1} + \mathbf{W}_{ox} \mathbf{x}_t + \mathbf{b}_o) \tag{5}$$
$$\tilde{\mathbf{c}}_t = \tanh(\mathbf{W}_{ch} \mathbf{h}_{t-1} + \mathbf{W}_{cx} \mathbf{x}_t + \mathbf{b}_c) \tag{6}$$
$$\mathbf{c}_t = \mathbf{f}_t \odot \mathbf{c}_{t-1} + \mathbf{i}_t \odot \tilde{\mathbf{c}}_t \tag{7}$$
$$\mathbf{h}_t = \mathbf{o}_t \odot \tanh(\mathbf{c}_t) \tag{8}$$

Here $\sigma(\cdot)$ denotes the element-wise sigmoid function and $\odot$ denotes element-wise multiplication. $\mathbf{f}_t$, $\mathbf{i}_t$, and $\mathbf{o}_t$ are referred as the forget, input, and output gates, which can be interpreted as controlling how much we reset, write to, and read from the memory cell $\mathbf{c}_t$. LSTM has better gradient properties than simple RNNs (see Figure 2) because of the mechanism in Equation 7, which introduces a path between $\mathbf{c}_{t-1}$ and $\mathbf{c}_t$ which is modulated only by the forget gate. We also remark that gated recurrent units (Cho et al., 2014) alleviate the vanishing gradient problem using this exact same idea.

NARX RNNs (Lin et al., 1996) also address the vanishing gradient problem, but using a mechanism that is orthogonal to (and possibly complementary to) that of LSTM. This is done by allowing *delays*, or direct connections from the past. NARX RNNs in their general form are described by

$$\mathbf{h}_t = f(\mathbf{h}_{t-1}, \mathbf{h}_{t-2}, \ldots, \mathbf{x}_t, \mathbf{x}_{t-1}, \ldots, \boldsymbol{\theta}) \tag{9}$$

---

[2]$\tanh$ is a common choice, but it is of course also possible to use other activation functions.

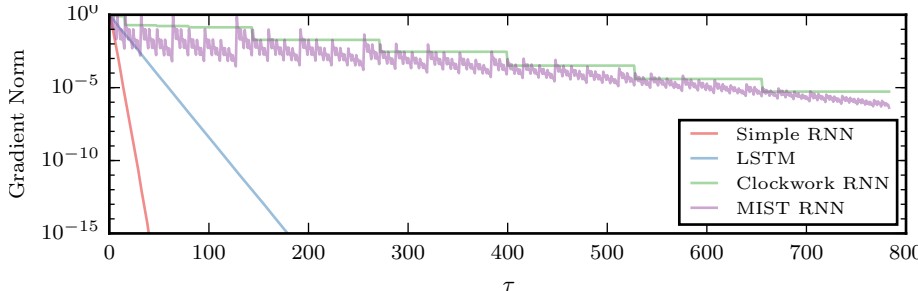

Figure 2: Gradient norms $\|\frac{\partial^+ l_t}{\partial \mathbf{h}_{t-\tau}}\|$ averaged over a batch of examples during permuted MNIST training. Unlike Clockwork RNNs and MIST RNNs, simple RNNs and LSTM capture essentially no learning signal from inputs that are far from the loss.

but literature typically assumes the specific variant explored in (Lin et al., 1996),

$$\mathbf{h}_t = \tanh\left(\left[\sum_{d=1}^{n_d} \mathbf{W}_d \mathbf{h}_{t-d}\right] + \mathbf{W}_x \mathbf{x}_t + \mathbf{b}\right) \tag{10}$$

which we refer to as *simple* NARX RNNs.

Note that simple NARX RNNs require approximately $n_d$ as much computation and $n_d$ as many parameters as their simple-RNN counterpart (with $n_d = 1$), which greatly hinders their applicability in practice. To our knowledge, this drawback holds for all NARX RNN variants before MIST RNNs. For example, in (Soltani & Jiang, 2016), higher-order recurrent neural networks (HORNNs) are defined precisely as simple NARX RNNs, and every variant in the paper suffers from this exact same problem. And, in (Zhang et al., 2016), a simple NARX RNN architecture is defined that is limited to having precisely two delays with non-zero weights. This way, at the expense of having fewer, longer paths to the past, parameter and computation counts are only doubled.

The previous work that is most similar to ours is that of Clockwork RNNs (Koutnik et al., 2014), which split weights and hidden units into partitions, each with a distinct period. When it's not a partition's time to tick, its hidden units are passed through unchanged, and so Clockwork RNNs in some ways mimic NARX RNNs. However Clockwork RNNs differ in two key ways. First, Clockwork RNNs sever high-frequency-to-low-frequency paths, thus making it difficult to learn long-term behavior that must be detected at high frequency (for example, learning to depend on quick motions from the past for activity recognition). Second, Clockwork RNNs require hidden units to be partitioned *a priori*, which in practice is difficult to do in any meaningful way. NARX RNNs (and in particular MIST RNNs) suffer from neither of these drawbacks.

Many other approaches have also been proposed to capture long-term dependencies. Notable approaches include maintaining a generative model over inputs and learning to process only unexpected inputs (Schmidhuber, 1992), operating explicitly at multiple time scales (El Hihi & Bengio, 1995), Hessian-free optimization (Martens & Sutskever, 2011), using associative or explicit memory (Plate, 1993; Danihelka et al., 2016; Graves et al., 2014; Weston et al., 2015), and initializing or restricting weight matrices to be orthogonal (Arjovsky et al., 2016; Henaff et al., 2016).

## 3 THE VANISHING GRADIENT PROBLEM IN THE CONTEXT OF NARX RNNs

In (Bengio et al., 1994; Pascanu et al., 2013), gradient decompositions and sufficient conditions for vanishing gradients are presented for simple RNNs, which contain one path between times $t - \tau$ and $t$. Here, we use the *chain rule for ordered derivatives* (Werbos, 1990) to connect gradient components to paths and edges, which in turn provides a simple extension of the results from (Bengio et al., 1994; Pascanu et al., 2013) to general NARX RNNs. We remark that we rely on slightly overloaded notation for clarity, as otherwise notation becomes cumbersome (see (Werbos, 1989)).

We begin by disambiguating notation, as the symbol $\frac{\partial \mathbf{f}}{\partial \mathbf{x}}$ is routinely overloaded in literature. Consider the Jacobian of $\mathbf{f}(\mathbf{x}, \mathbf{u}(\mathbf{x}))$ with respect to $\mathbf{x}$. We let $\frac{\partial^+ \mathbf{f}}{\partial \mathbf{x}}$ denote $\frac{\partial \mathbf{f}(\mathbf{x})}{\partial \mathbf{x}}$, a collection of full

derivatives, and we let $\frac{\partial \mathbf{f}}{\partial \mathbf{x}}$ denote $\frac{\partial \mathbf{f}(\mathbf{x}, \mathbf{u})}{\partial \mathbf{x}}$, a collection of partial derivatives. This lets us write the ordinary chain rule as $\frac{\partial^+ \mathbf{f}}{\partial \mathbf{x}} = \frac{\partial \mathbf{f}}{\partial \mathbf{x}} \frac{\partial^+ \mathbf{x}}{\partial \mathbf{x}} + \frac{\partial \mathbf{f}}{\partial \mathbf{u}} \frac{\partial^+ \mathbf{u}}{\partial \mathbf{x}}$. Note that this notation is consistent with (Werbos, 1989; 1990), but is the exact opposite of the convention used in (Pascanu et al., 2013).

## 3.1 THE CHAIN RULE FOR ORDERED DERIVATIVES

Consider an ordered system of $n$ vectors $\mathbf{v}_1, \mathbf{v}_2, \ldots, \mathbf{v}_n$, where each is a function of all previous:

$$\mathbf{v}_i \equiv \mathbf{v}_i(\mathbf{v}_{i-1}, \mathbf{v}_{i-2}, \ldots, \mathbf{v}_1), \quad 1 \le i \le n \tag{11}$$

The chain rule for ordered derivatives expresses the full derivatives $\frac{\partial^+ \mathbf{v}_i}{\partial \mathbf{v}_j}$ for any $j < i$ in terms of the full derivatives that relate $\mathbf{v}_i$ to all previous $\mathbf{v}_k$:

$$\frac{\partial^+ \mathbf{v}_i}{\partial \mathbf{v}_j} = \sum_{i \ge k > j} \frac{\partial^+ \mathbf{v}_i}{\partial \mathbf{v}_k} \frac{\partial \mathbf{v}_k}{\partial \mathbf{v}_j}, \quad j < i \tag{12}$$

## 3.2 GRADIENT DECOMPOSITION FOR GENERAL NARX RNNS

Consider NARX RNNs in their general form (Equation 9), which we remark encompasses other RNNs such as LSTM as special cases. Also, for simplicity, consider the situation that is most often encountered in practice, where the loss at time $t$ is defined in terms of the current state $\mathbf{h}_t$ and its own parameters $\boldsymbol{\theta}_l$ (which are independent of $\boldsymbol{\theta}$).

$$l_t = f_l(\mathbf{h}_t, \boldsymbol{\theta}_l) \tag{13}$$

(This is in not necessary, but we proceed this way to make the connection with RNNs in practice evident. For example, $f_l$ may be a linear transformation with parameters $\boldsymbol{\theta}_l$ followed by squared-error loss.) Then the Jacobian (or transposed gradient) with respect to $\boldsymbol{\theta}$ can be written as

$$\frac{\partial^+ l_t}{\partial \boldsymbol{\theta}} = \frac{\partial f_l}{\partial \mathbf{h}_t} \frac{\partial^+ \mathbf{h}_t}{\partial \boldsymbol{\theta}} \tag{14}$$

because the additional term $\frac{\partial f_l}{\partial \boldsymbol{\theta}_l} \frac{\partial^+ \boldsymbol{\theta}_l}{\partial \boldsymbol{\theta}}$ is $\mathbf{0}$. Now, by letting $\mathbf{v}_1 = \boldsymbol{\theta}$, $\mathbf{v}_2 = \mathbf{x}_1$, $\mathbf{v}_3 = \mathbf{x}_2$, and so on in Equations 11 and 12, we immediately obtain

$$\frac{\partial^+ \mathbf{h}_t}{\partial \boldsymbol{\theta}} = \sum_{\tau=0}^{t-1} \frac{\partial^+ \mathbf{h}_t}{\partial \mathbf{h}_{t-\tau}} \frac{\partial \mathbf{h}_{t-\tau}}{\partial \boldsymbol{\theta}} \tag{15}$$

because all of the partials $\frac{\partial \mathbf{x}_{t-\tau}}{\partial \boldsymbol{\theta}}$ are $\mathbf{0}$.

Equations 14 and 15 extend Equations 3 and 4 of (Pascanu et al., 2013) to general NARX RNNs, which encompass simple RNNs, LSTM, etc., as special cases. This decomposition breaks $\frac{\partial^+ \mathbf{h}_t}{\partial \boldsymbol{\theta}}$ into its temporal components, making it clear that the spectral norm of $\frac{\partial^+ \mathbf{h}_t}{\partial \mathbf{h}_{t-\tau}}$ plays a major role in how $\mathbf{h}_{t-\tau}$ affects the final gradient $\frac{\partial^+ l_t}{\partial \boldsymbol{\theta}}^T$. In particular, if the norm of $\frac{\partial^+ \mathbf{h}_t}{\partial \mathbf{h}_{t-\tau}}$ is extremely small, then $\mathbf{h}_{t-\tau}$ has only a negligible effect on the final gradient, which in turn makes it extremely difficult to learn from events that occurred at $t - \tau$.

## 3.3 CONNECTING GRADIENT COMPONENTS TO PATHS AND EDGES

Equations 14 and 15, along with the chain rule for ordered derivatives, let us connect gradient components to paths and edges, which is useful for a) gaining insights into various architectures and b) solidifying intuitions from backpropagation through time which suggest that short paths between $t - \tau$ and $t$ facilitate gradient flow. Here we provide an overview of the main idea; please see the appendix for a full derivation.

By applying the chain rule for ordered derivatives to expand $\frac{\partial^+ \mathbf{h}_t}{\partial \mathbf{h}_{t-\tau}}$ in Equation 15, we obtain a sum over $\tau$ terms. However each term involves a partial derivative between $\mathbf{h}_t$ and a prior hidden state, and thus all of these terms are $\mathbf{0}$ with the exception of those states that share an edge with $\mathbf{h}_t$. Now,

for each term, we can repeat this process. This then yields non-zero terms only for hidden states which can be connected to $\mathbf{h}_t$ through two edges. We can then continue to apply the chain rule for ordered derivatives repeatedly, until only partial derivatives remain.

Upon completion, we have a sum over gradient components, with each component corresponding to exactly one path from $t - \tau$ to $t$ and being a product over its path's edges. The spectral norm corresponding to any particular path $(t - \tau \to t' \to t'' \to \cdots \to t)$ can then bounded as

$$\left\| \frac{\partial \mathbf{h}_t}{\partial \mathbf{h}_{t''\cdots}} \cdots \frac{\partial \mathbf{h}_{t'}}{\partial \mathbf{h}_{t-\tau}} \right\| \leq \left\| \frac{\partial \mathbf{h}_t}{\partial \mathbf{h}_{t''\cdots}} \right\| \cdots \left\| \frac{\partial \mathbf{h}_{t'}}{\partial \mathbf{h}_{t-\tau}} \right\| \leq \lambda^{n_e} \tag{16}$$

where $\lambda$ is the maximum spectral norm of any factor and $n_e$ is the number of edges on the path. Terms with $\lambda < 1$ diminish exponentially fast, and when all $\lambda < 1$, shortest paths dominate[3].

## 4    MIXED HISTORY RECURRENT NEURAL NETWORKS

Viewing gradient components as paths, with each component being a product with one factor per edge along the path, gives us useful insight into various RNN architectures. When relating a loss at time $t$ to events at time $t - \tau$, simple RNNs and LSTM contain shortest paths of length $\tau$, while simple NARX RNNs contain shortest paths of length $\tau/n_d$, where $n_d$ is the number of delays.

One can envision many NARX RNN architectures with non-contiguous delays that reduce these shortest paths further. In this section we introduce one such architecture using base-2 exponential delays. In this case, for all $\tau \leq 2^{n_d-1}$, shortest paths exist with only $\log_2 \tau$ edges; and for $\tau > 2^{n_d-1}$, shortest paths exist with only $\tau/2^{n_d-1}$ edges (see Figure 1). Finally we must avoid the parameter and computation growth of simple NARX RNNs. We achieve this by sharing weights over delays, instead using an attention-like mechanism (Bahdanau et al., 2015) over delays and a reset mechanism from gated recurrent units (Cho et al., 2014).

The proposed architecture, which we call mixed history RNNs (MIST RNNs), is described by

$$\mathbf{a}_t = \text{softmax}(\mathbf{W}_{ah}\mathbf{h}_{t-1} + \mathbf{W}_{ax}\mathbf{x}_t + \mathbf{b}_a) \tag{17}$$

$$\mathbf{r}_t = \sigma(\mathbf{W}_{rh}\mathbf{h}_{t-1} + \mathbf{W}_{rx}\mathbf{x}_t + \mathbf{b}_r) \tag{18}$$

$$\mathbf{h}_t = \tanh\left(\mathbf{W}_h\left[\mathbf{r}_t \odot \sum_{i=0}^{n_d-1} a_{ti}\mathbf{h}_{t-2^i}\right] + \mathbf{W}_x\mathbf{x}_t + \mathbf{b}\right) \tag{19}$$

Here, $\mathbf{a}_t$ is a learned vector of $n_d$ convex-combination coefficients and $\mathbf{r}_t$ is a reset gate. At each time step, a convex combination of delayed states is formed according to $\mathbf{a}_t$; units of this combination are reset according to $\mathbf{r}_t$; and finally the typical linear layer and nonlinearity are applied.

## 5    EXPERIMENTS

Here we compare MIST RNNs to simple RNNs, LSTM, and Clockwork RNNs. We begin with the sequential permuted MNIST task and the copy problem, synthetic tasks that were introduced to explicitly test RNNs for their ability to learn long-term dependencies (Hochreiter & Schmidhuber, 1997; Martens & Sutskever, 2011; Le et al., 2015; Arjovsky et al., 2016; Henaff et al., 2016; Danihelka et al., 2016). Next we move on to 3 tasks for which it is plausible that very long-term dependencies play a role: recognizing surgical maneuvers from robot kinematics, recognizing phonemes from speech, and classifying activities from smartphone motion data. We note that for all architectures involved, many variations can be applied (variational dropout, layer normalization, zoneout, etc.). We keep experiments manageable by comparing architectures without such variations.

### 5.1    SEQUENTIAL PMNIST CLASSIFICATION

The sequential MNIST task (Le et al., 2015) consists of classifying 28x28 MNIST images (LeCun et al., 1998) as one of 10 digits, by scanning pixel by pixel – left to right, top to bottom – and emitting

---

[3]We remark that it is also possible for gradient contributions to explode exponentially fast, however this problem can be remedied in practice with gradient clipping. None of the architectures discussed in this work, including LSTM, address the exploding gradient problem.

Table 1: Test-set error rates for sequential pMNIST classification. Hidden unit counts ($n_h$) vary to match parameter counts with LSTM (approx. 42,000 parameters), except models marked with $+$ (which have more parameters). $\alpha^*$ denotes the optimal learning rate according to validation error.

|  | $n_h$ | $\log_{10} \alpha^*$ | Error Rate (%) |
| --- | --- | --- | --- |
| Simple RNNs | 198 | $-2.27 \pm 0.10$ | $12.9 \pm 0.8$ |
| LSTM | 100 | $-1.11 \pm 0.11$ | $10.4 \pm 0.7$ |
| Clockwork RNNs | 256 | $-1.91 \pm 0.23$ | $15.7 \pm 1.2$ |
| MIST RNNs | 139 | $-1.35 \pm 0.08$ | $5.5 \pm 0.2$ |
| LSTM$^+$ | 139 | $-0.90 \pm 0.26$ | $8.8 \pm 0.6$ |
| LSTM$^+$ | 512 | $-1.08 \pm 0.18$ | $7.6 \pm 0.7$ |
| MIST RNNs$^+$ | 512 | $-1.19 \pm 0.13$ | $4.5 \pm 0.1$ |

a label upon completion. Sequential pMNIST (Le et al., 2015) is a challenging variant where a random permutation of pixels is chosen and applied to all images before classification. LSTM with 100 hidden units is used as a baseline, with hidden unit counts for other architectures chosen to match the number of parameters. Means and standard deviations are computed using the top 5 randomized trials out of 50 (ranked according to performance on the validation set), with random learning rates and initializations. Additional experimental details can be found in the appendix.

Test error rates are shown in Table 1. Here, MIST RNNs outperform simple RNNs, LSTM, and Clockwork RNNs by a large margin. We remark that our LSTM error rates are consistent with best previously-reported values, such as the error rates of 9.8% in (Cooijmans et al., 2016) and 12% in (Arjovsky et al., 2016), which also use 100 hidden units. One may also wonder if the difference in performance is due to hidden-unit counts. To test this we also increased the LSTM hidden unit count to 139 (to match MIST RNNs), and continued to increase the capacity of each model further. MIST RNNs significantly outperform LSTM in all cases.

We also used this task to visualize gradient magnitudes as a function of $\tau$ (the distance from the loss which occurs at time $t = 784$). Gradient norms for all methods were averaged over a batch of 100 random examples early in training; see Figure 2. Here we can see that simple RNNs and LSTM capture essentially no learning signal from steps that are far from the loss. To validate this claim further, we repeated the 512-unit LSTM and MIST RNN experiments, but using only the last 200 permuted pixels (rather than all 784). LSTM performance remains the same (7.4% error, within 1 standard deviation) whereas MIST RNN performance drops by 15 standard deviations (6.0% error).

## 5.2 THE COPY PROBLEM

The copy problem is a synthetic task that explicitly challenges a network to store and reproduce information from the past. Our setup follows (Arjovsky et al., 2016), which is in turn based on

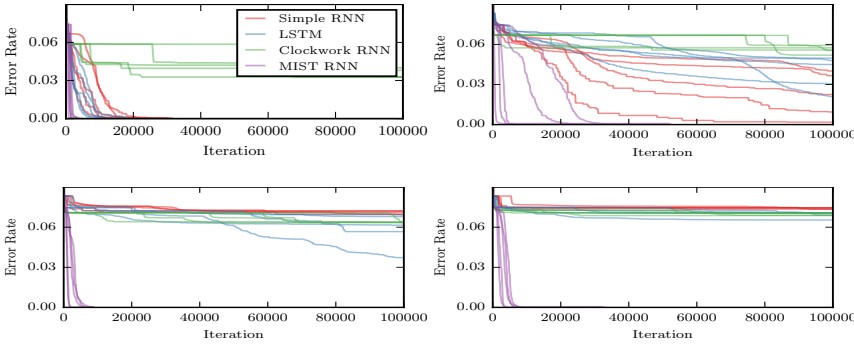

Figure 3: Validation curves for the copy problem with copy delays of 50 (upper left), 100 (upper right), 200 (lower left), and 400 (lower right).

Table 2: Error rates for surgical maneuver recognition. Hyperparameters were copied from (DiPietro et al., 2016), where they were tuned for LSTM with peephole connections. Our LSTM does not include peephole connections (see Section 2).

|  | $n_h$ | Error Rate (%) |
|---|---|---|
| LSTM (DiPietro et al., 2016) | 1024 | $12.2 \pm 2.7$ |
| Simple RNNs | 1024 | $38.0 \pm 6.2$ |
| LSTM | 1024 | $13.9 \pm 3.0$ |
| Clockwork RNNs | 1024 | $22.5 \pm 5.0$ |
| MIST RNNs | 1024 | $14.1 \pm 3.9$ |

(Hochreiter & Schmidhuber, 1997). An input sequence begins with $L$ relevant symbols to be copied, is followed by a delay of $D - 1$ special blank symbols and 1 special go symbol, and ends with $L$ additional blank symbols. The corresponding target sequence begins with $L + D$ blank symbols and ends with a copy of the relevant symbols from the inputs (in the same order). We run experiments with copy delays of $D = 50, 100, 200,$ and $400$. LSTM with 100 hidden units is used as a baseline, with hidden unit counts for other architectures chosen to match the number of parameters. Additional experimental details can be found in the appendix.

Results are shown in Figure 3, showing validation curves of the top 5 randomized trials out of 50, with random learning rates and initializations. With a short copy delay of $D = 50$, we can see that all methods other than Clockwork RNNs can solve the task in a reasonable amount of time. However, as the copy delay $D$ is increased, we can see that simple RNNs and LSTM become unable to learn a solution, whereas MIST RNNs are relatively unaffected. We also note that our LSTM results are consistent with those in (Arjovsky et al., 2016; Henaff et al., 2016).

Note that Clockwork RNNs are expected to fail for large delays (for example, the second symbol can only be seen by the highest-frequency partition, so learning to copy this symbol will fail for precisely the same reason that simple RNNs fail). However, here they also fail for short delays, which is surprising because the high-speed partition resembles a simple RNN. We hypothesized that this failure is due to hidden unit counts / parameter counts: here, the high-frequency partition is allocated only 256 / 8 = 32 hidden units. To test this hypothesis, we reran the Clockwork RNN experiments with 1024 hidden units, so that 128 are allocated to the high-frequency partition. Indeed, under this configuration (with 10x as many parameters), Clockwork RNNs do solve the task for a delay of $D = 50$ and fail to solve the task for all higher delays, thus behaving like simple RNNs.

## 5.3 SURGICAL MANEUVER RECOGNITION

Here we consider the task of online surgical maneuver recognition using the MISTIC-SL dataset (Gao et al., 2014; DiPietro et al., 2016). Maneuvers are fairly long, high-level activities; examples include *suture throw* and *knot tying*. The dataset was collected using a *da Vinci*, and the goal is to map robot kinematics over time (e.g., $x$, $y$, $z$) to gestures over time (which are densely labeled as 1 of 4 maneuvers on a per-frame basis). We follow (DiPietro et al., 2016), which achieves state-of-the-art performance on this task, as closely as possible, using the same kinematic inputs, test setup, and hyperparameters; details can be found in the original work or in the appendix. The primary difference is that we replace their LSTM layer with our layers. Results are shown in Table 2. Here MIST RNNs match LSTM performance (with half the number of parameters).

## 5.4 PHONEME RECOGNITION

Here we consider the task of online framewise phoneme recognition using the TIMIT corpus (Garofolo et al., 1993). Each frame is originally labeled as 1 of 61 phonemes. We follow common practice and collapse these into a smaller set of 39 phonemes (Lee & Hon, 1989), and we include glottal stops to yield 40 classes in total. We follow (Greff et al., 2016) for data preprocessing and (Halberstadt, 1998) for training, validation, and test splits. LSTM with 100 hidden units is used as a baseline, with hidden unit counts for other architectures chosen to match the number of parameters. Means and standard deviations are computed using the top 5 randomized trials out of 50 (ranked according

Table 3: Test-set error rates for TIMIT phoneme recognition. Hidden unit counts ($n_h$) vary to match parameter counts with LSTM (approx. 44,000 parameters). $\alpha^*$ denotes the optimal learning rate according to validation error.

|  | $n_h$ | $\log_{10} \alpha^*$ | Error Rate (%) |
| --- | --- | --- | --- |
| Simple RNNs | 197 | $-1.09 \pm 0.25$ | $34.1 \pm 0.3$ |
| LSTM | 100 | $-0.63 \pm 0.06$ | $32.1 \pm 0.2$ |
| Clockwork RNNs | 248 | $-1.03 \pm 0.31$ | $38.2 \pm 0.4$ |
| MIST RNNs | 139 | $-0.91 \pm 0.16$ | $32.0 \pm 0.3$ |

Table 4: Test-set error rates for MobiAct activity classification. Hidden unit counts ($n_h$) vary to match parameter counts with LSTM (approx. 44,000 parameters), with the exception of LSTM$^+$ (approx. 88,000 parameters). $\alpha^*$ denotes the optimal learning rate according to validation error.

|  | $n_h$ | $\log_{10} \alpha^*$ | Error Rate (%) |
| --- | --- | --- | --- |
| Simple RNNs | 203 | $-1.91 \pm 0.18$ | $49.2 \pm 2.7$ |
| LSTM | 100 | $-0.89 \pm 0.12$ | $38.8 \pm 1.5$ |
| LSTM$^+$ | 141 | $-0.45 \pm 0.17$ | $37.8 \pm 2.1$ |
| Clockwork RNNs | 256 | $-1.09 \pm 0.31$ | $40.3 \pm 4.8$ |
| MIST RNNs | 141 | $-1.39 \pm 0.21$ | $29.0 \pm 5.2$ |

to performance on the validation set), with random learning rates and initializations. Other experimental details can be found in the appendix. Table 3 shows that LSTM and MIST RNNs perform nearly identically, which both outperform simple RNNs and Clockwork RNNs.

## 5.5 ACTIVITY RECOGNITION FROM SMARTPHONES

Here we consider the task of sequence classification from smartphones using the MobiAct (v2.0) dataset (Chatzaki et al., 2016). The goal is to classify each sequence as jogging, running, sitting down, etc., using smartphone motion data over time. Approximately 3,200 sequences were collected from 67 different subjects. We use the first 47 subjects for training, the next 10 for validation, and the final 10 for testing. Means and standard deviations are computed using the top 5 randomized trials out of 50 (ranked according to performance on the validation set), with random learning rates and initializations. Other experimental details can be found in the appendix. Results are shown in Table 4. Here, MIST RNNs outperform all other methods, including LSTM and LSTM$^+$, a variant with the same number of hidden units and twice as many parameters.

## 6 CONCLUSIONS AND FUTURE WORK

In this work we analyzed NARX RNNs and introduced a variant which we call MIST RNNs, which 1) exhibit superior vanishing-gradient properties in comparison to LSTM and previously-proposed NARX RNNs; 2) improve performance substantially over LSTM on tasks requiring very long-term dependencies; and 3) require even fewer parameters and computation than LSTM. One obvious direction for future work is the exploration of other NARX RNN architectures with non-contiguous delays. In addition, many recent techniques that have focused on LSTM are immediately transferable to NARX RNNs, such as variational dropout (Gal & Ghahramani, 2016), layer normalization (Ba et al., 2016), and zoneout (Krueger et al., 2016), and it will be interesting to see if such enhancements can improve MIST RNN performance further.

## ACKNOWLEDGMENTS

Removed for anonymity.

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

## 7 APPENDIX: GRADIENT COMPONENTS AS PATHS

Here we will apply Equation 12 repeatedly to associate gradient components with paths connecting $t - \tau$ to $t$, beginning with Equation 15 and handling simple RNNs and simple NARX RNNs in order. Applying Equation 12 to expand $\frac{\partial^+ \mathbf{h}_t}{\partial \mathbf{h}_{t-\tau}}$, we obtain

$$\frac{\partial^+ \mathbf{h}_t}{\partial \mathbf{h}_{t-\tau}} = \sum_{t \geq t' > t-\tau} \frac{\partial^+ \mathbf{h}_t}{\partial \mathbf{h}_{t'}} \frac{\partial \mathbf{h}_{t'}}{\partial \mathbf{h}_{t-\tau}} \tag{20}$$

### 7.0.1 SIMPLE RNNS

For simple RNNs, by examining Equation 2, we can immediately see that all partials $\frac{\partial \mathbf{h}_{t'}}{\partial \mathbf{h}_{t-\tau}}$ are $\mathbf{0}$ except for the one satisfying $t' = t - \tau + 1$. This yields

$$\frac{\partial^+ \mathbf{h}_t}{\partial \mathbf{h}_{t-\tau}} = \frac{\partial^+ \mathbf{h}_t}{\partial \mathbf{h}_{t-\tau+1}} \frac{\partial \mathbf{h}_{t-\tau+1}}{\partial \mathbf{h}_{t-\tau}} \tag{21}$$

Now, by applying Equation 12 again to $\frac{\partial^+ \mathbf{h}_t}{\partial \mathbf{h}_{t-\tau+1}}$, and then to $\frac{\partial^+ \mathbf{h}_t}{\partial \mathbf{h}_{t-\tau+2}}$, and so on, we trace out a path from $t - \tau$ to $t$, as shown in Figure 1, finally resulting the single term

$$\frac{\partial \mathbf{h}_t}{\partial \mathbf{h}_{t-1}} \cdots \frac{\partial \mathbf{h}_{t-\tau+2}}{\partial \mathbf{h}_{t-\tau+1}} \frac{\partial \mathbf{h}_{t-\tau+1}}{\partial \mathbf{h}_{t-\tau}} \tag{22}$$

which is associated with the *only* path from $t - \tau$ to $t$, with one factor for each edge that is encountered along the path.

### 7.0.2 SIMPLE NARX RNNS AND GENERAL NARX RNNS

Next we consider simple NARX RNNs, again by expanding Equation 15. From Equation 10, we can see that up to $n_d$ partials are now nonzero, and that any particular partial $\frac{\partial \mathbf{h}_{t'}}{\partial \mathbf{h}_{t-\tau}}$ is nonzero if and only if $t' > t - \tau$ and $t'$ and $t - \tau$ share an edge. Collecting these $t'$ as the set $V_{t-\tau} = \{t' : t' > t - \tau \text{ and } (t - \tau, t') \in E\}$, we can write

$$\frac{\partial^+ \mathbf{h}_t}{\partial \mathbf{h}_{t-\tau}} = \sum_{t' \in V_{t-\tau}} \frac{\partial^+ \mathbf{h}_t}{\partial \mathbf{h}_{t'}} \frac{\partial \mathbf{h}_{t'}}{\partial \mathbf{h}_{t-\tau}} \tag{23}$$

We can then apply this exact same process to each $\frac{\partial^+ \mathbf{h}_t}{\partial \mathbf{h}_{t'}}$; by defining $V_{t'} = \{t'' : t'' > t' \text{ and } (t', t'') \in E\}$ for all $t'$, we can write

$$\frac{\partial^+ \mathbf{h}_t}{\partial \mathbf{h}_{t-\tau}} = \sum_{t' \in V_{t-\tau}} \sum_{t'' \in V_{t'}} \frac{\partial^+ \mathbf{h}_t}{\partial \mathbf{h}_{t''}} \frac{\partial \mathbf{h}_{t''}}{\partial \mathbf{h}_{t'}} \frac{\partial \mathbf{h}_{t'}}{\partial \mathbf{h}_{t-\tau}} \tag{24}$$

By continuing this process until only partials remain, we obtain a summation over all possible paths from $t - \tau$ to $t$. *Each term* in the sum is a product over factors, one per edge:

$$\frac{\partial \mathbf{h}_t}{\partial \mathbf{h}_{t''' \cdots}} \cdots \frac{\partial \mathbf{h}_{t''}}{\partial \mathbf{h}_{t'}} \frac{\partial \mathbf{h}_{t'}}{\partial \mathbf{h}_{t-\tau}} \tag{25}$$

The analysis is nearly identical for general NARX RNNs, with the only difference being the specific sets of edges that are considered.

## 8 APPENDIX: EXPERIMENTAL DETAILS

### 8.1 GENERAL EXPERIMENTAL SETUP

Everything in this section holds for all experiments except surgical maneuver recognition, as in that case we mimicked DiPietro et al. (2016) as closely as possible, as described above.

All weight matrices are initialized using a normal distribution with a mean of 0 and a standard deviation of $1/\sqrt{n_h}$, where $n_h$ is the number of hidden units. All initial hidden states (for $t < 1$) are initialized to **0**. For optimization, gradients are computed using full backpropagation through time, and we use stochastic gradient descent with a momentum of 0.9, with gradient clipping as described by Pascanu et al. (2013) at 1, and with a minibatch size of 100. Biases are generally initialized to 0, but we follow best practice for LSTM by initializing the forget-gate bias to 1 Gers et al. (2000); Jozefowicz et al. (2015). For Clockwork RNNs, 8 exponential periods are used, as in the original paper. For MIST RNNs, 8 delays are used. We avoid manual learning-rate tuning in its entirety. Instead we run 50 trials for each experimental configuration. In each trial, the learning rate is drawn uniformly at random in log space between $10^{-4}$ and $10^1$, and initial weight matrices are also redrawn at random. We report results over the top 10% of trials according to validation-set error. (An alternative option is to report results over *all* trials. However, because the majority of trials yields bad performance for all methods, this simply blurs comparisons. See for example Figure 3 of Greff et al. (2016), which compares these two options.)

## 8.2 SEQUENTIAL pMNIST CLASSIFICATION

Data preprocessing is kept minimal, with each input image individually shifted and scaled to have mean 0 and variance 1. We split the official training set into two parts, the first 58,000 used for training and the last 2,000 used for validation. Our test set is the same as the official test set, consisting of 10,000 images. Training is carried out by minimizing cross-entropy loss.

## 8.3 COPY PROBLEM: EXPERIMENTAL DETAILS

In our experiments, the $L$ relevant symbols are drawn at random (with replacement) from the set $\{0, 1, \ldots, 9\}$; $D$ is always a multiple of 10; and $L$ is chosen to be $D/10$. This way the simplest baseline of always predicting the blank symbol yields a constant error rate for all experiments. No input preprocessing of any kind is performed. In each case, we generate 100,000 examples for training and 1,000 examples for validation. Training is carried out by minimizing cross-entropy loss.

## 8.4 SURGICAL ACTIVITY RECOGNITION: EXPERIMENTAL DETAILS

We use the same experimental setup as DiPietro et al. (2016), which currently holds state-of-the-art performance on these tasks. For kinematic inputs we use positions, velocities, and gripper angles for both hands. We also use their leave-one-user-out teset setup, with 8 users in the case of JIGSAWS and 15 users in the case of MISTIC-SL. Finally we use the same hyperparameters: 1 hidden layer of 1024 units; dropout with $p = 0.5$; 80 epochs of training with a learning rate of 1.0 for the first 40 epochs and having the learning rate every 5 epochs for the rest of training. As mentioned in the main paper, the primary difference is that we replaced their LSTM layer with our simple RNN, LSTM, or MIST RNN layer. Training is carried out by minimizing cross-entropy loss.

## 8.5 PHONEME RECOGNITION: EXPERIMENTAL DETAILS

We follow Greff et al. (2016) and extract 12 mel frequency cepstral coefficients plus energy every 10ms using 25ms Hamming windows and a pre-emphasis coefficient of 0.97. However we do not use derivatives, resulting in 13 inputs per frame. Each input sequence is individually shifted and scaled to have mean 0 and variance 1 over each dimension. We form our splits according to Halberstadt (1998), resulting in 3696 sequences for training, 400 sequences for validation, and 192 sequences for testing. Training is carried out by minimizing cross-entropy loss. Means and standard deviations are computed using the top 5 randomized trials out of 50 (ranked according to performance on the validation set).

## 8.6 ACTIVITY RECOGNITION FROM SMARTPHONES

In (Chatzaki et al., 2016), emphasis was placed on hand-crafted features, and each subject was included during both training and testing (with no official test set defined). We instead operate on

the raw sequence data, with no preprocessing other than sequence-wise centering and scaling of inputs, and we define train, val, test splits so that subjects are disjoint among the three groups.

