# OpenReview forum: "Analyzing and Exploiting NARX Recurrent Neural Networks for Long-Term Dependencies"
_ICLR.cc/2018/Conference — Invite to Workshop Track_

### Official Review · AnonReviewer3 · 2017-11-26
**little novelty and unconvincing**

**Rating:** 3
**Confidence:** 4

**Review:**

The followings are my main critics of the paper:
1. Analysis does not provide any new insights.
2. Similar work (recurrent skip coefficient and the corresponding architecture in [1]) has been done, but has not been mentioned.
3. The experimental results are not convincing. This includes 1. the choices of tasks are limited -- very small in size, 2. the performance in pMNIST is worse than [1], under the same settings.

Hence I think the novelty of the paper is very little, and the experiments are not convincing.

[1] Architectural Complexity Measures of Recurrent Neural Networks. Saizheng Zhang, Yuhuai Wu, Tong Che, Zhouhan Lin, Roland Memisevic, Ruslan Salakhutdinov, Yoshua Bengio. NIPS, 2016.

---

> ### Author Response · Authors · 2018-01-04
> **Response to AnonReviewer3**
>
> Thank you for your review. We kindly note that some of the comments in this review are incorrect, and as such we sincerely hope that you are willing to reconsider your evaluation of our work.
>
> >>>>> The experimental results are not convincing. This includes 1. the choices of tasks are limited -- very small in size, 2. the performance in pMNIST is worse than [1], under the same settings.
>
> Point 2:
>
> Please note that this is incorrect. In [1], the best reported error rate for pMNIST is 6.0% error, whereas we obtain 5.5 +- 0.2% error. Also, their results (Table 2) correspond to a hyperparameter sweep, with s = 11 achieving 6.0% error. We require no such sweeps: our delays were kept fixed for all 5 tasks in the paper (still outperforming every model proposed in [1]).
>
> Point 1:
>
> Please note that we evaluated these methods across
>
> - 2 synthetic tasks that have been widely used for testing long-term dependencies, as was highlighted in Section 5 with references (Hochreiter et al., 1997; Martens et al., 2011; Le et al., 2015; Arjovsky et al., 2016; Henaff et al., 2016; Danihelka et al., 2016)
>
> - 3 real tasks that were chosen because they a) likely require long-term dependencies and b) are of moderate size so that statistically-significant results can be obtained.
>
> We followed the experimental design of [2], which also includes 3 real tasks of moderate size, preferring random hyperparameter sweeps and statistically-significant results over manual sweeps and statistically-questionable results. Also, please note that this design seems to be reasonable to the community, as [2] has been cited 400+ times since 2014.
>
> Regarding the dataset sizes: TIMIT is standard, with splits identical to [2]. MobiAct contains approximately 3200 sequences of mobile sensor data from 67 users, very similar in size to the datasets in [2]. MISTIC-SL is smaller in size, but we chose this task because long-term dependencies are required and because state of the art is held by LSTM (which we ended up matching with MIST RNNs).
>
> [1] Zhang et al. Architectural complexity measures of recurrent neural networks. Advances in neural information processing systems (NIPS), 2016.
>
> [2] Greff et al. LSTM: A search space odyssey. IEEE Trans. on Neural Networks and Learning Systems, 2016.
>
> >>>>> Similar work (recurrent skip coefficient and the corresponding architecture in [1]) has been done, but has not been mentioned.
>
> Based on this comment, we have added a discussion of [1] to the Background section. However kindly note that
>
> - with regard to the architecture, [1] proposes precisely a simple NARX RNN ([19], discussed extensively in our paper) with non-zero weights for only two delays. This bears little resemblance to our work. Most importantly, MIST RNNs provide exponentially-short paths to the past while maintaining fewer parameters and computations than LSTM. In contrast, [1] does not provide exponentially-short paths, and uses two delays to avoid high parameter/computation counts. In case there is any doubt about this, we quote [1]: "By using this specific construction, the recurrent skip coefficient increases from 1 (i.e., baseline) to k and the new model with extra connection has 2 hidden matrices (one from t to t + 1 and the other from t to t + k)."
>
> - with regard to skip coefficients, [1] defines a *measure* of shortest paths called Recurrent Skip Coefficients. However in [1] the motivation for this definition is "it is known that adding skip connections across multiple time steps may help improve the performance on long-term dependency problems [19, 20]." Again, [19] introduced simple NARX RNNs, as discussed extensively in our paper. Thus the extent to which [1]'s skip coefficients overlap with our work is that we both recognize that short paths are important. A difference between our work and [1] is that we provide a self-contained derivation of this.
>
> [1] Zhang et al. Architectural complexity measures of recurrent neural networks. Advances in neural information processing systems (NIPS), 2016.
>
> [19] Lin et al. Learning long-term dependencies in NARX recurrent neural networks. IEEE Transactions on Neural Networks, 7(6):1329–1338, 1996.
>
> [20] Sutskever et al. Temporal-kernel recurrent neural networks. Neural Networks, 23(2):239–243, 2010.
>
> >>>>> Analysis does not provide any new insights.
>
> The connection of gradient components to paths via the chain rule for ordered derivatives is new. However we agree that the analysis portion of the paper is not revolutionary - this was not the goal of the analysis. Our goals were to provide a self-contained justification of our approach and to extend the results from ([1], [2]) to general NARX RNNs.
>
> [1] Bengio et al. Learning long-term dependencies with gradient descent is difficult. IEEE Transactions on Neural Networks, 5(2):157-166, 1994.
>
> [2] Pascanu et al. On the difficulty of training recurrent neural networks. International Conference on Machine Learning (ICML), 28:1310-1318, 2013.

---

### Official Review · AnonReviewer2 · 2017-11-26
**The paper introduces a variant of the well-known (but as of today not very frequently used) NARX architecture for Recurrent Neural Networks. It is demonstrated that with the proposed method (MIST RNNs), good performance is achieved on several common RNN problems.**

**Rating:** 7
**Confidence:** 5

**Review:**

The presented MIST architecture certainly has got its merits, but in my opinion is not very novel, given the fact that NARX RNNs have been described 20 years ago, and Clockwork RNNs (which, as the authors point out in section 2, have a similar structure) have also been in use for several years. Still, the presented results are good, with standard LSTMs being substantially outperformed in three out of five standard RNN/LSTM benchmark tasks. The analysis in section 3 is decent (see however the minor comments below), but does not offer revolutionary new insights - it's perhaps more like a corollary of previous work (Pascanu et al., 2013).

Regarding the concrete results, I would have wished for a more detailed analysis of the more surprising results, in particular, for the copy task (section 5.2): Is it really true that Clockwork RNNs fail because they make it "difficult to learn long-term behavior that must be detected at high frequency" [section 2]? How relevant are the results in figure 2 (yes, the gradient properties are very different, but is this an issue for accuracy)? In the sequential pMNIST classification, what about increasing the LSTM number of hidden units? If this brings the error rate further down, one could ask why exactly the LSTM captures long-term structure so differently with different number of units?

In summary, for me this paper is solid, and although the architecture is not that new, it is worth bringing it again into the focus of attention.


Minor comments:
- In several places, the formulas are rather strange and/or occasionally incorrect. In particular,
* on the right-hand sind of the inline formula in section 3.1, the symbol v is missing completely, which cannot be right;
* in formula 16, the primes seem to be misplaced, and the symbols t', t''', etc. should be defined;
* the \theta_l in the beginning of section 3.3 (formula 13) is completely superfluous.
- The position of the tables and figures is rather weird, making the paper less readable than necessary. The authors should consider moving floating parts around (one could also move figure three to the bottom of a suitable page, for example).
- It is a matter of taste, but since all experimental results except the ones on the copy task are tabulated, one could think of adding a table with the results now contained in figure 3.

Relation to prior work: the authors are aware of most relevant work.

On p2 they write: "Many other approaches have also been proposed to capture long-term dependencies." There is one that seems close to what the authors do:

J. Schmidhuber. Learning complex, extended sequences using the principle of history compression. Neural Computation, 4(2):234-242, 1992

It is related to clockwork RNNs, about which the authors write:

"A recent architecture that is similar in spirit to our work is that of Clockwork RNNs (Koutnik et al., 2014), which split weights and hidden units into partitions, each with a distinct period. When it’s not a partition’s time to tick, its hidden units are passed through unchanged, thus in some ways mimicking the behavior of NARX RNNs. However Clockwork RNNs differ in two key ways. First, Clockwork RNNs sever high-frequency-to-low-frequency paths, thus making it difficult to learn long-term behavior that must be detected at high frequency (for example, learning to depend on quick motions from the past for activity recognition). Second, Clockwork RNNs require hidden units to be partitioned a priori, which in practice is difficult to do in any meaningful way. NARX RNNs suffer from neither of these drawbacks."

The neural history compressor, however, adapts to the frequency of unexpected events, by ticking only when there is an unpredictable event, thus overcoming some of the issues above. Perhaps this trick could further improve the system of the authors, as well as the Clockwork RNNs, at least for certain tasks?

General recommendation: Accept, provided the comments are taken into account.

---

> ### Author Response · Authors · 2018-01-04
> **Response to AnonReviewer2**
>
> We are pleased that you enjoyed our work. Thank you very much for your detailed review and insightful comments. We have done our best to address every question raised, and we have updated the paper to reflect every response here:
>
> >>>>> for the copy task (section 5.2): Is it really true that Clockwork RNNs fail because they make it "difficult to learn long-term behavior that must be detected at high frequency" [section 2]?
>
> For large delays (D >= 100), this is precisely the reason that Clockwork RNNs fail, but we see no way of providing further empirical evidence of this. We instead describe in detail why Clockwork RNNs must fail:
>
> - Symbol 0 can be 'copied ahead' by all partitions, and so perhaps it is possible to learn to replicate this symbol later in time.
>
> - Symbol 1 can only be seen by the highest-frequency partition (period of T = 1) because 1 % T = 0 for T = 1, but not T = 2, 4, 8, 16, etc. Also, this partition cannot send information to lower-frequency partitions. Hence Clockwork RNNs cannot learn to replicate symbol 1 for the exact same reason that a simple RNN cannot: the shortest past to the loss has at least D matrix multiplies and nonlinearities.
>
> - Symbol 2 can similarly only be seen by the two highest-frequency partitions (T = 1, T = 2), so we have a shortest path with D / 2 nonlinearities and matrix multiplies (a negligible difference for medium-to-large delays).
>
> - Symbol 3 can only be seen by the single highest-frequency partition because again 3 % T = 0 only for T = 1, so the situation is identical to symbol 1.
>
> - And so on. Hence Clockwork RNNs must fail to learn to copy most of these symbols for medium-to-large delays.
>
> For small delays (D = 50), Clockwork RNNs should solve the copy task, because the highest-frequency partition resembles a simple RNN. However, this partition has only 256 / 8 = 32 hidden units. We thus ran additional Clockwork RNN experiments with 1024 hidden units (and 10x as many parameters), with 128 units allocated to the high-frequency partition. We then see that Clockwork RNNs do solve the copy problem with a delay of 50 and continue to fail to solve the problem for higher delays, as expected.
>
> >>>>> In the sequential pMNIST classification, what about increasing the LSTM number of hidden units? If this brings the error rate further down, one could ask why exactly the LSTM captures long-term structure so differently with different number of units?
>
> We ran additional experiments with 512 units for both LSTM and MIST RNNs. LSTM obtains an improved error rate of 7.6%, and MIST RNNs obtain an improved error rate of 4.5%. However, we verified that capacity does not help with long-term dependencies; please see the next question.
>
> >>>>> How relevant are the results in figure 2 (yes, the gradient properties are very different, but is this an issue for accuracy)?
>
> We included Figure 2 to show that empirical observations match our expectations for gradient decay. To provide further empirical validation, we ran additional pMNIST experiments for the 512-unit LSTM and MIST RNNs:
>
> - Based on Figure 2, we used only the last 200 pixels (rather than all 784).
>
> - LSTM performance remained the same (within 1 std. dev., 7.4% error), showing that LSTM gained nothing from including the distant past.
>
> - MIST RNN performance degraded by 15 standard deviations (6.0% error), showing that MIST RNNs do benefit from the distant past.
>
> - Finally we note that MIST RNNs still outperform LSTM. This is expected since LSTM has trouble learning even from steps <= 200 from the loss (as shown in Fig. 2).
>
> >>>>> on the right-hand side of the inline formula in section 3.1, the symbol v is missing
>
> Thank you. This arose from merging two previous examples. Fixed.
>
> >>>>> in formula 16, the primes seem to be misplaced, and the symbols t', t''', etc. should be defined
>
> Fixed
>
> >>>>> the \theta_l in the beginning of section 3.3 (formula 13) is completely superfluous.
>
> We agree but include this to make the connection to practice immediately evident. We added a sentence to clarify this.
>
> >>>>> The position of the tables and figures is rather weird...
>
> Fixed.
>
> >>>>> Relation to prior work: the authors are aware of most relevant work... There is one that seems close to what the authors do: J. Schmidhuber. Learning complex, extended sequences using the principle of history compression. Neural Computation, 4(2):234-242, 1992 ...
>
> Learning a generative model over inputs to identify surprising inputs for processing is an interesting approach; we added this to the Background section.
>
> >>>>> Perhaps this trick could further improve the system of the authors, as well as the Clockwork RNNs, at least for certain tasks?
>
> We would not be surprised at all if this method can improve results for some tasks, especially those with highly-correlated, low-dimensional inputs such as MNIST (or even pMNIST). However, addressing this question fully would be far from trivial, so we leave it as future work.

---

### Official Review · AnonReviewer1 · 2017-11-27

**Rating:** 6
**Confidence:** 4

**Review:**

Summary: The authors introduce a variant of NARX RNNs, which has an additional attention mechanism and a reset mechanism. The attention is only applied on subsets of hidden states, referred as delays. The delays are aggregated into a vector using the attention coefficients as weights, and then this vector is multiplied by the reset gates.

The model sounds a bit incremental, however, the performance improvements over pMNIST, copy and MobiAct tasks are interesting.

A similar kind of architecture has been already proposed:
[1] Soltani et al. “Higher Order Recurrent Neural Networks”, arXiv 1605.00064

---

> ### Author Response · Authors · 2018-01-04
> **Response to AnonReviewer1**
>
> Thank you for your review. We also found it interesting that MIST RNNs can capture such long-term dependencies.
>
> >>>>> A similar kind of architecture has been already proposed: [1] Soltani et al. “Higher Order Recurrent Neural Networks”, arXiv 1605.00064
>
> Based on this comment, we have added a short discussion of [1] to the Background section.
>
> However, we would like to kindly note that [1] defines a "higher order recurrent neural network (HORNN)" precisely as a simple NARX RNN, which was introduced 20 years earlier in [2], and which was already discussed extensively in our paper.
>
> Importantly, every HORNN variant in [1] suffers from the same issue that is mentioned in our paper for simple NARX RNNs: the vanishing gradient problem is only mitigated mildly as n_d, the number of delays, increases; and simultaneously parameter and computation counts grow by this same factor n_d. We would like to emphasize that MIST RNNs are the first NARX RNNs that resolve both of these issues, by providing exponentially short connections to the past while maintaining even fewer parameters and computations than LSTM.
>
> [1] Rohollah Soltani and Hui Jiang. Higher order recurrent neural networks. arXiv preprint arXiv:1605.00064, 2016.
>
> [2] Tsungnan Lin, Bill G Horne, Peter Tino, and C Lee Giles. Learning long-term dependencies in NARX recurrent neural networks. IEEE Transactions on Neural Networks, 7(6):1329–1338, 1996.

---

### Author Response · Authors · 2018-01-04
**Added revision incorporating reviewer feedback**

Changes:

- The last 3 paragraphs of Section 2 (Background) were expanded and edited based on feedback from all 3 reviewers.

- Section 3 (The Vanishing Gradient Problem in the Context of NARX RNNs) was edited for clarity and to fix typos spotted by AnonReviewer2.

- Section 5.1 (Permuted MNIST results) was heavily modified based on AnonReviewer2's feedback. In particular, results were added with additional hidden-unit counts, and results were added to show that LSTM performance does not depend at all on information from the distant past (whereas MIST RNN performance does).

- A paragraph was added to the end of Section 5.2 (Copy Problem results) based on AnonReviewer2's feedback. In particular we discuss additional Clockwork RNN results; the reasons that Clockwork RNNs must fail for large delays; and show that Clockwork RNNs do indeed behave like simple RNNs if enough hidden units are provided.

- Figures and Tables were moved around for clarity, based on AnonReviewer2's feedback.

- Small miscellaneous edits were made throughout to open space for the previous changes.

---

### Decision · Program_Chairs · 2018-01-29
**ICLR 2018 Conference Acceptance Decision**

**Decision:**

Invite to Workshop Track

**Comment:**

I think the model itself is not very novel, as pointed by the reviewers and the analysis is not very insightful either. However, the results themselves are interesting and quite good (on the copy task and pMnist, but not so much the other datasets presented (timit etc) where it not clear that long term dependencies would lead to better results). Since the method itself is not very novel, the onus is upon the authors to make a strong case for the merits of the paper --  It would be worth exploring these architectures further to see if there are useful elements for real world tasks -- more so than is demonstrated in the paper --  for example showing it on tasks such as machine translation or language modelling tasks requiring long term propagation of information or even real speech recognition, not just basic TIMIT phone frame classification rate.

As a result, while I think the paper could make for an interesting contribution, in its present form, I have settled on recommending the paper for the workshop track.


As a side note, paper is related to paper 874 in that an attention model is used to look at the past. The difference is in how the past is connected to the current model.